# BK Polyomavirus in Pediatric Renal Transplantation—What We Know and What We Do Not

**DOI:** 10.3390/biomedicines12051093

**Published:** 2024-05-15

**Authors:** Benedetta Chiodini, Pauline Guillaume-Gentil, Charlotte Vanhomwegen, Elise Hennaut, Ksenija Lolin, Nathalie Tram, Alain Le Moine, Khalid Ismaili

**Affiliations:** 1Department of Pediatric Nephrology, Hôpital Universitaire de Bruxelles-HUDERF (HUB-HUDERF), Université Libre de Bruxelles (ULB), 1050 Brussels, Belgium; 2Department of Nephrology, Hôpital Universitaire de Bruxelles-Erasme (HUB-Erasme), European Plotkin Institute for Vaccinology, Université Libre de Bruxelles (ULB), 1050 Brussels, Belgium

**Keywords:** BK polyomavirus, BKPyV nephropathy, kidney transplantation

## Abstract

BK polyomavirus (BKPyV) is still a real threat in the management of kidney transplantation. Immunosuppressive treatment disrupts the equilibrium between virus replication and immune response, and uncontrolled BKPyV replication leads to nephropathy (BKPyV nephropathy). The first evidence of BKPyV reactivation in transplant recipients is the detection of viral shedding in urine, which appears in 20% to 60% of patients, followed by BKPyV viremia in 10–20% of kidney transplant recipients. BKPyV nephropathy eventually occurs in 1–10% of this population, mainly within the first 2 years post-transplantation, causing graft loss in about half of those patients. Few data exist regarding the pediatric population and we focus on them. In this paper, we review the existing diagnostic methods and summarize the evidence on the role of BKPyV humoral and cellular immunity in modulating the clinical course of BKPyV infection and as potential predictors of the outcome. We look at the known risk factors for BKPyV nephropathy in the immunosuppressed patient. Finally, we propose a sensible clinical attitude in order to screen and manage BKPyV infection in kidney transplant children.

## 1. Introduction

*Polyomaviridae* is a family of small, non-enveloped viruses with circular dsDNA genomes of approximately 5 kbp. This family includes four genera whose members have restricted host ranges, infecting almost exclusively mammals and birds [1]. The first polyomavirus was identified in 1953 as a filterable tumor-causing agent in mice, followed by Simian vacuolating virus (SV40) isolated from rhesus monkey kidney cells that had been used for poliovirus vaccine preparation in the 1960s [2]. The first two human polyomavirus species, both named after the index case patients, were identified in 1971 independently in two different patients. BK polyomavirus (BKPyV) was isolated from a urine sample of a renal transplant patient [3], and JC virus was isolated from the brain tissue of a patient with progressive multifocal leukoencephalopathy [4]. BK and JC viruses have been the only well-known human polyomaviruses for decades. However, more than 50 years of research have provided crucial insights into the cellular and molecular biology of these viruses, and up to now, at least 13 human polyomavirus species have been identified [1,5].

Primary BKPyV infection is mainly acquired early in life and is most likely transmitted horizontally by direct contact, aerosol, or fecal–oral routes [6]. Approximately 80% of the general population has detectable antibodies to BKPyV, which appear at a point in childhood and persist throughout life [7]. The mechanisms of persistence and reactivation of the virus remain poorly understood. The BKPyV is believed to first replicate in the respiratory tract and then spread to other organs through the bloodstream, notably renal tubular epithelial and urothelial cells [8]. BKPyV remains clinically latent in immunocompetent hosts, but after kidney transplantation, immunosuppressive treatment disrupts the equilibrium between virus replication and immune response, resulting in uncontrolled BKPyV replication. BKPyV is currently considered by nephrologists a real threat in the management of kidney transplantation, as it may lead to the graft being lost.

This review focuses on the existing diagnostic methods, the evidence on the role of BKPyV humoral and cellular immunity in the modulation of the clinical course of BKPyV infection, and the potential predictors of the outcome. The characteristics of the pediatric population are emphasized, and a sensible clinical attitude in order to screen and manage BKPyV infection in kidney transplant children is proposed.

## 2. Kidney Graft Involvement

In kidney transplant recipients, BKPyV replication starts in the distal tubular epithelial cells, leading to necrosis and the initiation of local damage and inflammation. The spread of virus in the adjacent environment will result in viruria, which is the first evidence of BKPyV reactivation and appears in 20% to 60% of patients [9]. After this initial stage, disruption of the tubular basement membrane can occur, leading to BKPyV viremia in 10 to 20% of kidney transplant recipients. BKPyV nephropathy, which is caused by the recruitment of inflammatory cells in the tubule-interstitial space and viral spreading to proximal cells, occurs in 1–10% of renal allograft recipients. It appears mainly within the first 2 years post-transplantation and causes graft loss in about half of those patients [10,11].

The virological and immunological determinants of the progression from self-limited BKPyV viruria or viremia to BKPyV nephropathy in kidney transplant recipients are poorly understood. As for the source of BKPyV, the donor origin has been repeatedly suggested, although virus reactivation in recipients can also occur, and both sources may not be mutually exclusive [12,13,14,15].

## 3. Immune Response

A review paper published by Lamarche and colleagues has extensively illustrated the virus-intrinsic features, the post-transplant microenvironment, as well as the host immune factors that underlie the pathophysiology of BKPyV-associated nephropathy [15].

### 3.1. Humoral Immune Response

The role of BKPyV humoral immunity in modulating of the clinical course of BKPyV infection is still debated. In 1983, Rosen et al. reported a severe case of end-stage renal disease due to a BKPyV infection in a six-year-old immunocompromised boy with congenital dysgammaglobulinemia [16]. Renal biopsy at the time of his presentation revealed tubulo-interstitial lesions with mononuclear infiltrates and numerous BKPyV inclusions. A hypothesis was raised that humoral immune deficiency was responsible for the severity of the BKPyV infection in this child.

Pre-transplant serologies for cytomegalovirus and Epstein–Barr virus in donor and recipient are now routinely performed with the aim of estimating the risk of infection and planning the prophylactic treatment. Similarly, could pre-transplant serology for BKPyV be used as a screening tool to evaluate the individual risk for BKPyV infection?

Serotypes I and IV represent more than 90% of the strains found in kidney transplant patients. Studies have suggested that high BKPyV-specific antibody titers against the donor strain before transplantation might have a protective role. Wunderink et al. found that a higher level of serotype I antibodies before transplantation was strongly correlated with a lower risk of BKPyV infection [17]. A study by Solis et al. in 2018 revealed the key role of the anti-BKPyV genotype–specific neutralizing antibody titers in protecting against BKPyV infection [18]. Recently, Dakroub et al. used a commercial antibody for an enzyme-linked immunosorbent assay (ELISA) in order to assess the BKPyV serostatus of 329 adult kidney transplant recipients and 222 matched donors [19]. Their results showed an increased risk of developing post-transplantation BKPyV viremia from 4.3% for the D−/R+ group to 12.1% for the D+/R+ group, climbing to 37.5% for the D+/R− group (*p* < 0.05) [19].

However, several authors estimate that the humoral immune response does not play a major role in the control of the BKPyV virulence [9,20,21,22,23,24].

### 3.2. Cellular Immune Response

Already 40 years ago, Drummond et al. showed that the BKPyV-specific cell-mediated immune response was the main mechanism in controlling viral replication [25]. This finding was further supported by a study on the cellular immune response to the JC virus in patients with progressive multifocal leukoencephalopathy. In those patients, JC virus-specific cytotoxic T lymphocytes were found to be a key factor in containing the disease [26].

Impairment of cell-mediated immune response allows BKPyV reactivation, which begins with the active viral replication in the graft, followed by viral shedding in the urine, and eventually viremia, nephropathy, and, in the worst-case scenario, graft loss [22,27]. In kidney transplant recipients, low levels of BKPyV-specific interferon gamma-producing T cells correlate with progression to BKPyV nephropathy [28].

Currently, the plasma BKPyV-DNA load is used after kidney transplantation to identify patients at risk of BKPyV nephropathy. However, some authors advocate that BKPyV viremia alone is not an accurate predictor of the clinical outcome and that more reliable parameters are needed to distinguish patients with self-limiting viremia from those with a higher risk of BKPyV nephropathy [29]. Therefore, monitoring pre- and post-transplant BKPyV-specific T cells was suggested as a potential accurate marker to identify those kidney transplant recipients at increased risk.

Data on children are very scarce. As for the adult population, in a recent pediatric study, high levels of BKPyV-specific CD4 and/or CD8 T-cells were found in cases of self-limiting viremia, whereas a lack or low level of BKPyV-specific T cells was associated with long-term viremia and florid BKPyV nephropathy [30].

## 4. Diagnosis

### 4.1. Urine Cytology

Decoy cells are virally infected uro-epithelial cells characterized on light microscopy by a ground-glass appearance with intra-nuclear inclusions. Their presence in urine has been used for decades as a marker for BKPyV infection, although their positive predictive value is weak and limited by the high intra-observer variability [10]. Moreover, urine decoy cells are not specific for BKPyV infection, having also been described in adenovirus and cytomegalovirus infections [31]. However, many institutions continue to screen renal transplant recipients through the detection of decoy cells, and they measure BKPyV-DNA in the plasma only if decoy cells persist. The main reason is that in many countries, the public health system and/or insurance policies do not cover the cost of PCR testing [32].

### 4.2. BKPyV Viruria

Urine BKPyV-PCR has been proposed as a more reliable screening tool for renal transplant recipients, as this method could detect the infection at an early stage. Thresholds used for the risk of viremia and BKPyV nephropathy are commonly >10^7^ copies/mL [33]. Adult data shown that a BKPyV viruria was associated with BKPyV nephropathy in only one-third of cases, while no patients with a viruria under this cut-off developed the disease [34]. Therefore, this test is only useful when it is negative, as it makes the diagnosis of BKPyV nephropathy improbable.

Few data concern the pediatric population. In the CERTAIN Registry study, about one quarter of children developed a high level BKPyV viruria within the first post-transplantation year, while only 3% ended with a BKPyV nephropathy [35]. In our ongoing Belgian study on 55 transplanted children between 2010 and 2022, 18% developed a high level BKPyV viruria already within the first 6 months post-transplantation, while during the same early period, 6% progressed towards a high-level viremia and none to BKPyV nephropathy (Data presented at the 37th congress of the French Pediatric Nephrology Society, Nice 15–17 November 2023).

### 4.3. BKPyV Viremia

Regular screening for BKPyV replication by plasma viral load is now universally recommended in kidney transplant recipients [36,37]. A recent Belgian study with systematic biopsy and BKPyV-DNA load follow-up has shown a very high negative predictive value (0.989) when the plasma viral load was below 10^4^ copies/mL. Performing a biopsy for confirmation of BKPyV nephropathy is therefore not necessary in cases of low viral load [38].

Conversely, persistent high-level BKPyV viremia, (>10^4^ copies/mL) is associated with an increased risk of BKPyV nephropathy, and it currently even defines a presumptive BKPyV nephropathy [9,38,39]. Therefore, in this scenario, guidelines universally recommend a reduction in maintenance immunosuppression [39].

As regards as the children population, in the CERTAIN Registry study 14% of patients developed a high-level viremia (presumptive BKPyV nephropathy) within the first year and needed a therapeutic intervention [35]. However, many reports have observed self-limiting BKPyV viremia in kidney recipients without any immunosuppression reduction [39,40,41]. Therefore, in these cases, pre-emptive reduction and/or modification of immunosuppressive drugs could not only be unnecessary but potentially at increased risk of rejection.

### 4.4. BKPyV-Specific Cell Immune Monitoring

BKPyV-specific cell-mediated immune response has been a recent area of research, as BKPyV-specific cellular immunity seems to play an important role in controlling viral replication. Adult and pediatric studies observed that an increase in BKPyV-specific T cells was correlated with the viral clearance in kidney transplant recipients [28,30]. BKPyV-specific cell immune monitoring has therefore been suggested as a prognostic tool to identify patients who are at risk of BKPyV nephropathy [41].

Recently, Ahlenstiel-Grunow and colleagues performed a detailed analysis of the BKPyV-specific cellular immune response in a German cohort of transplanted children at the onset of BKPyV viremia [30]. They found that a lack or low levels of BKPyV-specific CD4 and CD8 T cells were associated with an increased risk of persistent viremia, florid BKPyV nephropathy, and the urgent need for a reduction in immunosuppression. Whereas patients with adequate BKPyV-specific CD4 T cells (≥0.5 cells/µL) and/or CD8 T cells (≥0.1 cells/µL) showed transient, self-limiting viremia without BKPyV-associated complications.

Although the analysis of BKPyV-specific CD4 and CD8 T cells might offer an accurate prognostic tool in order to tailor the level of immunosuppression in kidney transplant recipients, up to now, the price, availability, and technical hitches make these tests impractical for use in routine care.

### 4.5. Allograft Biopsy

Standardized graft biopsy using immunohistochemistry (SV40 T antigen staining or in situ hybridization) remains the gold standard to confirm BKPyV nephropathy [36,37]. The Banff working group on BKPyV nephropathy classified the disease into three grades based on histology to allow comparative analyses and improvement in predicting clinical presentation and outcome [42]. Typical findings are focal interstitial mononuclear inflammatory cell infiltrates, necrotic tubular epithelium, and the presence of homogenous intra-nuclear inclusions. Tubular atrophy and interstitial fibrosis remain the most important predictors of poor outcomes [43].

In the CERTAIN Registry study, biopsy-proven BKPyV nephropathy was diagnosed in 4.5% of children during a 5-year post-transplantation period [35]. Within the 14 patients with biopsy-proven BKPyV nephropathy and after therapeutically center-dependent management, 4 children (29%) showed a renal function recovery, 8 children a stabilization (57%), and 2 patients (14%) ended with a deterioration of the kidney function and a graft loss, respectively [35].

Unlike in adults, more than 20% of these patients developed BKPyV nephropathy beyond the second year post-transplant [35]. Although current guidelines recommend screening for BKPyV replication only once per year after the second year post-transplant, these pediatric observations suggest a more vigilant approach.

Negative biopsy results do not necessarily rule out BKPyV nephropathy, as the disease can be focal at the start, and sampling errors are always possible [44]. Two cores containing medulla are required for an adequate biopsy sample [37]. Therefore, in patients with a negative kidney biopsy and a BKPyV viremia (>10^4^ copies/mL) for more than 4 weeks, the diagnosis of “presumptive BKPyV nephropathy” should be retained [36].

## 5. Risk Factors of BKPyV Nephropathy in the Immunosuppressed Patient

It is widely agreed that the introduction of more potent immunosuppression, both for the induction and the maintenance phase, has increased the risk of BKPyV nephropathy in transplanted kidneys [30,35,45].

### 5.1. Immunosuppression

***Thymoglobulin (ATG):*** As for the induction therapy, ATG has been clearly incriminated when compared to no induction or induction with IL-2 receptor blockers [9,45].

***Tacrolimus (TAC):*** Results from the large Organ Procurement and Transplantation Network (OPTN) Registry found an association between the TAC-based maintenance regimens and a higher incidence of treatment for BKPyV nephropathy within the first 2 years post-transplantation, and this when compared to cyclosporine (CsA)-based regimens [45]. TAC has also been implicated in more recent prospective studies on the adult population [46,47].

The pediatric cohort analysis of the United Network for Organ Sharing (UNOS) database on children followed for 5 years post-transplantation found that TAC therapy at the time of discharge was significantly associated with graft failure due to BKPyV nephropathy as compared to graft failure due to other causes [48]. The CERTAIN Registry study also found that TAC-based immunosuppression was independently associated with presumed BKPyV nephropathy in transplanted children [35].

***Mycophenolate mofetil (MMF):*** The OPTN registry study found an association between the MMF-based regimen and a higher incidence of treatment for BKPyV nephropathy within the first 2 years post-transplantation when compared to the azathioprine-based regimen [45].

Not surprisingly, the combination TAC-MMF has been associated with a particularly higher risk of BKPyV infection [9,45], even in children [35].

***Inhibitors of mammalian targets of rapamycin (mTORi):*** Studies have found that mTORi have an antiviral effect without losing their immunosuppressive action against graft rejection [49]. In vitro experiments using BKPyV-infected renal epithelial cell lines found that sirolimus reduces BKPyV large T-antigen expression and its genome replication by blocking the protein-kinase Akt/mTOR pathway [50,51]. In clinics, an important role of mTORi in decreasing BKPyV infection has been observed [45,52], especially when combined with the low-dose calcineurin inhibitor CNI [52,53,54,55,56]. Up to now, no reliable data exist on transplanted children.

### 5.2. Other Risk Factors

In addition to immunosuppression, other independent risk factors for BKPyV infection have been suggested, mainly recipient characteristics such as male gender [46], young [45] and advanced age [46], obstructive uropathy [35], and prior graft loss due to BKPyV nephropathy [29]. Also, specific graft characteristics have been suspected, such as organs from BKPyV seropositive donors to seronegative recipients, the number of HLA mismatches, ischemic injury, and ureteral stent placement at transplantation [20,35].

Transplanted children have an increased risk of developing a BKPyV serious infection. The possible explanation is that those patients have not been previously exposed to the virus, and when immunosuppressed, they acquire a primary and more severe infection [35,45]. On the contrary, the increased rate of BKPyV nephropathy in the elderly is plausible because of the reduction with increasing age in global humoral and cellular immunity to viruses [46]. It is therefore strongly recommended that there are more frequent surveillance in these two extreme-age populations.

## 6. How to Manage BKPyV in Pediatric Kidney Transplantation

### 6.1. BKPyV Screening

Our own experience on 55 transplanted children between 2010 and 2022 showed that 12% of patients progressed towards a high-level viremia (presumptive BKPyV nephropathy) within the first 2 post-transplant years, with 50% of them already affected within the first 6 months and nearly 85% within the first year [Data presented at the 37th congress of the French Pediatric Nephrology Society, Nice 15–17 November 2023].

Therefore, we suggest that kidney transplanted children should be screened for plasma BKPyV-DNA monthly the first year, then every 3 months the second year, and afterward every 6 months.

If BKPyV viremia load is >10^3^ copies/mL in two measurements at less than 3 weeks of interval, BKPyV load should be monitored for potential infection with frequent plasma sampling, at least every two weeks. If the viremia load is >10^4^ copies/mL in two measurements, presumptive BKPyV nephropathy should be considered even in the absence of renal biopsy, and the first step of management should be taken [38] (Figure 1).

### 6.2. Immunosuppression Management

The first step in the management of presumptive or proven BKPyV nephropathy is reducing maintenance immunosuppression as CNI and/or MMF. TAC trough levels are commonly targeted at <6 ng/mL, cyclosporine to <150 ng/mL, and MMF equal to or less than half the daily maintenance dose. During a reduction in immunosuppressive therapy, plasma BKPyV-DNA should be tested every 2–3 weeks, as should the levels of BKPyV-specific T cells when possible.

If necessary, a next step could be switching to mTORi plus low-dose CNI after cessation of MMF [57]. Unfortunately, this aggressive reduction in immunosuppression may inevitably lead to acute rejection and/or graft loss [58].

### 6.3. Adjunctive Therapies

In patients with persistent BKPyV nephropathy, despite adequately reduced immunosuppression, the use of adjunctive therapies may be considered [29,36,37]. However, no randomized controlled trials could provide evidence that their use alone or in combination, is superior to the reduction in immunosuppression alone [29].

***Intravenous immunoglobulin (IVIG):*** IVIG could play a broad anti-inflammatory, immunomodulatory role or directly neutralize viral activity. Clinical studies showing clearance of plasma BKPyV-DNA following IVIG administration do exist [59], though their real efficacy is difficult to evaluate as other antiviral interventions are always performed concomitantly [29]. IVIG has been administered in doses ranging from 0.1 to 2 g/Kg in conjunction with reduced immunosuppression.

***Cidofovir:*** Cidofovir is a nucleoside analog of cytosine, which has an antiviral activity inhibiting the DNA polymerase. Although randomized clinical trials assessing its efficacy are pending, a few adult studies did report a stabilization of renal function and/or clearance of viremia [60]. In addition, several side effects have been reported, including nephrotoxicity, myelosuppression, nausea, diarrhea, and anterior uveitis [29]. To our best knowledge, no robust pediatric studies have been published up to now [61].

***Leflunomide:*** Leflunomide has unique antiviral and immunosuppressive properties, inhibiting pyrimidin synthesis and protein kinase activity. It has been orally administered as a replacement for discontinued MMF, with a loading dose of 100 mg for 3 to 5 days, followed by a maintenance dose of 20 to 60 mg daily [62]. Significant toxic effects have been reported, including hepatitis, hemolysis, thrombotic microangiopathy, and bone marrow suppression [9,29]. In children, very scarce case reports reporting the efficacy of Leflunomide have been published [63,64].

In order to avoid rejection, the return to routine maintenance immunosuppression after successful clearance of BKPyV viremia should be considered under careful monitoring of plasma viral loads. In case a rejection occurs, it should be treated according to standard protocol.

## 7. Conclusions

Due to the limited number of pediatric studies, the diagnosis and treatment of BKPyV infections vary considerably between centers and countries [65]. The vast majority of children are managed according to adult guidelines. However, findings obtained in adult renal transplant recipients cannot necessarily be extrapolated to the pediatric patient population. Most of the pediatric recipients may present at the time of transplant a BKPyV serostatus D+/R− and are therefore at higher risk of developing an early and serious BKPyV infection. We have nevertheless attempted in the following flow chart to summarize a sensible clinical attitude in order to screen and manage BKPyV infection in kidney transplant children (Figure 1).

## Figures and Tables

**Figure 1 biomedicines-12-01093-f001:**
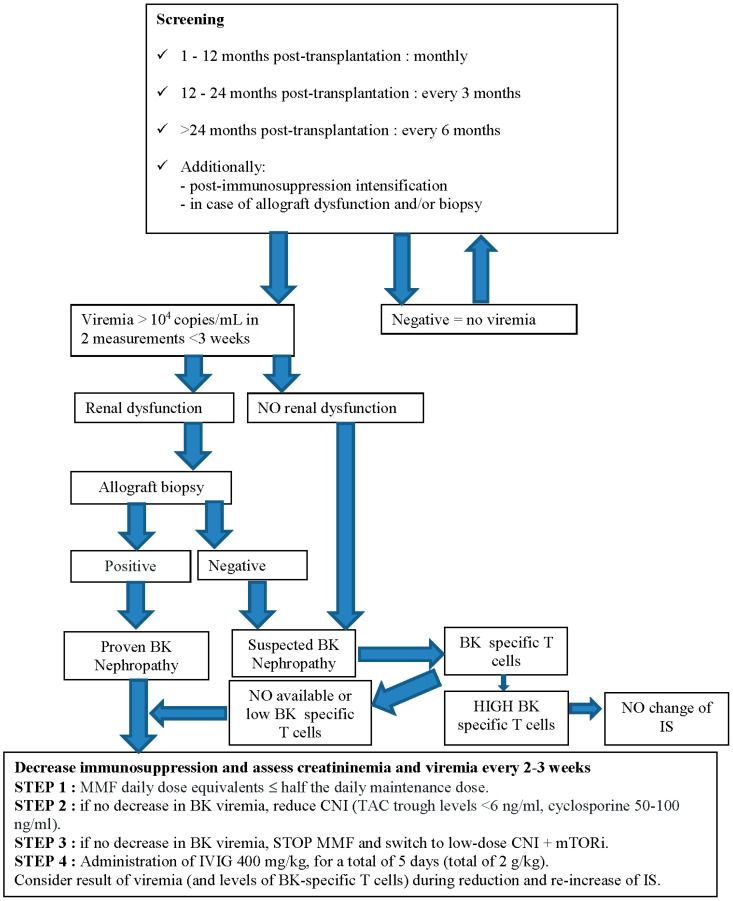
Proposed flow chart of BK virus screening and treatment strategy.

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
