# Peer review of "BK Polyomavirus in Pediatric Renal Transplantation—What We Know and What We Do Not"

_biomedicines, 2024, doi:10.3390/biomedicines12051093_

Round 1
Reviewer 1 Report
Comments and Suggestions for Authors
I considered the manuscript entitled “BK polyomavirus in pediatric renal transplantation. What we know and what we don’t.” By Benedetta Chiodini, et al, which is intended to be published in Biomedicines journal.
The study is well aimed, exposed, and discussed. However, scarce new information is given. The unique interest is that it deals with pediatric population where there is less information. However, few differences with adult population
Author Response
Dear Reviewer,
thank you for your comment.
As you said data and evidences on the BKPyV nephropathy within the pediatric population are scarce and the aim of our paper is to summarize what is known and still unclear. Moreover, we propose a flow chart for the screening and management of BKPyV infection in kidney transplant children.

Reviewer 2 Report
Comments and Suggestions for Authors
The author chose timely topic for this review since limited data exists for paediatric cases. The paper reviews BK polyomavirus kidney transplant infection and consecutive kidney transplant management, available diagnostic methods, as well as the role of immune response in infection progression. Additionally risk factors and clinical approaches for BKPyV treatment in paediatric kidney transplant recipients were explored. The review is clearly written. However, I have some suggestions for improvement.
1. Figure 1 need to presented without red lines and signs of word editing tools. some part of the text were interpolated in the reference list.
2. Additional figure would be beneficial to graphically clarify the immune response humeral vs cellular
3. Additional paragraph in relation to clinical prognosis in accordance to data from available clinical cases or papers.
4. spell check and grammar (some fonts are not the same)
5. reference list is not in accordance to journal guidelines
6. in the affiliations there is no postal code, and to information about corresponding author
Author Response
1. Figure 1 need to presented without red lines and signs of word editing tools. some part of the text were interpolated in the reference list. The figure 1 has been reformatted
2. Additional figure would be beneficial to graphically clarify the immune response humeral vs cellular. The aim of this review was more to deal with the clinical aspects of the BKPyV nephropathy than to deep into the immune mechanisms involved in the pathophysiology. However, at the beginning of the Immune Response paragraph, we cited an extensive illustrated review paper on the immunopathology of BKPyV in renal transplantation (now ref. 15)
3. Additional paragraph in relation to clinical prognosis in accordance to data from available clinical cases or papers.
- At the end of the paragraph on the Humoral Immune Response, we added the following sentences and data:
"Their results showed an increased risk of developing post-transplantation BKPyV viremia from 4.3% for the D-/R+ group to 12.1 % for the D+/R+ group, climbing to 37.5% for the D+/R- group (P<0.05) [19]."
- At the paragraph 4.5 Allograft biopsy, we added the following sentence and data:
"Within the 14 patients with a biopsy-proven BKPyV nephropathy, and after a therapeutically center-dependent management, 4 children (29%) showed a renal function recovery, 8 children a stabilization (57%), and 2 patients (14%) ended with a deterioration of the kidney function and a graft loss, respectively [35]."
4. spell check and grammar (some fonts are not the same). The spelling, grammar and font have been checked
5. Reference list is not in accordance to journal guidelines. We have adapted the References to the journal guidelines
6. in the affiliations there is no postal code, and to information about corresponding author. The informations about the Corresponding Author have been added to the first page
Reviewer 3 Report
Comments and Suggestions for Authors
Dear Authors!
Thank you for the opportunity to review your manuscript.
Polyomavirus BK is an important problem in nephrology, including pediatric and in kidney transplant. The PVBK is one of the reasons of nephropathy and could lead to graft loss in the infected people. The data about the pathogenesis and immune system functioning is crucial to diagnostics, treatment and outcomes in people, underwent kidney transplant. The manuscript is a structured review with different subheadings focuses on the immune system response. Some parts of the review looks alike short, e.g. 2 and 3 and could be elaborated. The treatment part contains the information of the treatment and manuscript has a figure, summarizes BK virus diagnostics and treatment.
The conclusion is relatively short and might be elaborated too.
Author Response
Dear reviewer, thank you for your comments!
In accordance with your suggestions, the Paragraph 1 and 2 have been merged. Paragraph 3 is now Paragraph 2 - Kidney Graft Involvement-. The pathophysiology stages have been explained and emphasized and a reference added (reference 15).
The Conclusion has been elaborated.
Reviewer 4 Report
Comments and Suggestions for Authors
Minor revisions:
1. Generally, an introduction section of a review paper provides some background information about the topic, outlines the scope of the study, and states the purpose or objectives of the review. It is advised that the introduction section be expanded, and more references added to state the abovementioned recommendations.
2. Even though there is a scientific value in the chosen subject, all sections appear to be very short and skimming through the subject. Considering that this topic is not widely discussed it becomes even more vital to provide all the existing information. It would be beneficial for the paper and scientific community to expand each section and add more detail information with added references.
3. The section 7. How to manage BKPyV in pediatric kidney transplantation in its current form is more suitable for a research paper or scientific report. Instead, a review paper synthesizes the results from several primary literature papers to produce a coherent argument about a topic or focused description of a field. It is recommended to re-write the section to synthesize or analyze research already conducted in primary sources.
4. Current review paper is following a slightly different format and has no discussion or conclusion sections, instead there is a very short section 8 Concluding remarks. True value in a good review paper lays not only in summarising existing current knowledge of the chosen subject, but also in providing a critical evaluation and new and fresh insights. Thus, traditional Discussion and Conclusion section may help to organize thoughts and guide the reader. Discussion section is intended to highlight the meaning, importance, and relevance of the work. It should focus on explaining and evaluating the information presented in the review and not merely presenting a synopsis of the previous chapters. It should also make an argument to support the conclusion. The conclusion provides a closure for the reader and reminds about the contents and importance of the research. It is thus recommended to expand the last section to account for the points above.
5. Kindly do the thorough editing of the review to address the issues of different fonts and extra spaces in front of the references.
6. Figure 1 appears to be unedited and should be replaced with an edited publication quality version.
Author Response
Dear Reviewer, thank you for your comments!
- The Introduction has been expanded and the aim of this review explained.
- The aim of this Paper was mainly to give an easy and straightforward review on this subject, trying to specially focus on the scarce pediatric evidences. In accordance to the reviewer suggestions, several Paragraphs have been re-elaborated and expanded (Introduction, Kidney Graft Involvement, Immune Response, Diagnosis, and Conclusion).
- and 4. The subject of the Management of BKPyV infection is extremely vast and controversial and not evidence-based. The aim of this paper is to give an insight of the most agreed and up to date clinical attitude. The Conclusion has been now elaborated.
5. and 6. We re-edited the entire paper, references and Figure 1